

**Cyclic activity of Fuego de Colima volcano (Mexico):**
**insights from satellite thermal data and non-linear models**
Silvia Massaro[1*], Antonio Costa[2], Roberto Sulpizio[1,3], Diego Coppola[4], Lucia Capra[5]
[1]Istituto per la Dinamica dei Processi Ambientali – Consiglio Nazionale delle Ricerche, Via R. Cozzi 53, 20125, Milan (Italy)
[2]Istituto Nazionale di Geofisica e Vulcanologia, Via D. Creti 12, 40128, Bologna (Italy)
[3]Dipartimento di Scienze della Terra e Geoambientali, Università di Bari, Via Orabona 4, 70125, Bari (Italy)
[4]Dipartimento di Scienze della Terra, Università di Torino, Via Valperga Caluso, 35, 10129, Turin (Italy)
[5]Centro de Geociencias UNAM, Campus Juriquilla, Queretaro (Mexico)
*Corresponding Author: Silvia Massaro (silvia-massaro@libero.it)

**Abstract**
The Fuego de Colima volcano (Mexico) showed a complex eruptive behaviour with periods of
rapid and slow lava dome growth, punctuated by explosive activity. We reconstructed the weekly
discharge rate average between 1998 and 2018 by means of satellite thermal data integrated with
published discharge rate data. By using spectral and wavelet analysis, we found a multi-year long-,
multi-month intermediate-, and multi-week short-term cyclic behaviour during the period of the
investigated eruptive activity, as those of many others dome-forming volcanoes. We use numerical
modelling in order to investigate the non-linear cyclic eruptive behaviour considering a magma
feeding system composed of a dual or a single magma chamber connected to the surface through an
elastic dyke evolving into a cylinder conduit in the shallowest part. We investigated the cases in
which the periodicity is controlled by i) the coupled deep-shallow magma reservoirs, ii) the single
shallow chamber, and iii) the elastic shallow dyke when is fed by a fixed influx rate or a constant
pressure. The model outputs indicate that the observed multi-year periodicity (1.5-2.5 years) can be
described by the fluctuations controlled by a shallow magma chamber with a volume of 20-50 km$^3$
coupled with a deep reservoir of 500 km$^3$, connected through a deep elastic dyke. The multi-month
periodicity (ca. 5 - 10 months) appears to be controlled by the shallow magma chamber for the
same range of volumes. The short-term multi-week periodicity (ca. 2.5 - 5 weeks) can be
reproduced considering a fixed influx rate or constant pressure at the base of the shallower dyke.
This work provides new insights on the non-linear cyclic behaviour of Fuego de Colima, and a
general framework for the comprehension of eruptive behaviour of andesitic volcanoes.





## 1. Introduction

Lava dome forming eruptions are relatively long-lived events, lasting from several months to several decades (e.g. Merapi, Indonesia, Siswowidjoyo et al., 1995; Kelut, Indonesia, De Bélizal et al., 2012; Fuego de Colima, Mexico, Lamb et al., 2014; Santiaguito, Guatemala, Harris et al., 2002), and usually punctuated by dome collapses and explosive (Vulcanian) episodes. Discharge rates can change widely over a range of time scales, reflecting the physical mechanisms involved in the transfer of magma to the Earth's surface (Melnik et al., 2008; Odbert and Wadge 2009). Dome growth is not a continuous process and periodic behaviour has been commonly observed at several volcanoes, including Santiaguito (Guatemala, Harris et al., 2003), Mt St Helens (USA, Swanson and Holcomb, 1990), and Soufrière Hills (Montserrat, Voight et al., 1998; Loughlin et al., 2010; Wadge et al., 2010; Nicholson et al., 2011). Periodic behaviours can be complex, showing systematic or non-systematic temporal changes as the eruption progresses (Denlinger and Hoblitt, 1999; Costa et al., 2007a; Melnik et al., 2008; Bernstein et al., 2013; Wolpert et al., 2016), and may be characterized by short-, intermediate- and long-term periodicities (Costa et al., 2007a; Melnik et al., 2008; Costa et al., 2012; 2013; Melnik and Costa, 2014). The short- and intermediate-term periodicities (hours or weeks) are generally explained by the upper conduit pressurization related to the non-linear ascent of magma flow (Denlinger and Hoblitt, 1999; Melnik and Sparks, 1999; Voight et al., 1999; Wylie et al., 1999; Ozerov et al., 2003; Lensky et al., 2004, Costa et al., 2007a,b; 2012; Kozono and Koyaguchi, 2012). This is because the lower part of the dyke-conduit acts as a capacitor that allows magma to be stored temporarily and released during the more intense phase of discharge (Costa et al., 2007a,b; Melnik et al., 2008; Costa et al. 2012; 2013). The long-term periodicity, with time scales from months to decades (Voight et al., 2000; Belousov et al., 2002; Sparks and Young, 2002; Wadge et al., 2006), is usually controlled by pressure variations in magma reservoirs (Barmin et al., 2002; Costa et al., 2007b; Melnik et al., 2008; Melnik and Costa, 2014). Since historical times, the Fuego de Colima volcano (Mexico; Fig.1a) has been characterised by decade-lasting cycles of dome growth alternating with Vulcanian explosions, ended with sub-





Plinian eruptions (the last two occurred in 1818 and 1913, (Luhr, 2002; Saucedo et al., 2005; Norini
et al., 2010; Heap et al., 2014; Massaro et al., 2018a). The most recent cycle started after the 1913
eruption, and it is characterized by lava domes extruded with minor seismicity at high magma
temperatures (960-1020°C; Savov et al., 2008). As for other dome eruptions (Sparks, 1997), dome
growth at Fuego de Colima can be explained by complex non-linear pressure variations during
magma ascent from magma reservoirs (e.g. Melnik and Costa, 2014), cooling, crystallization,
degassing (e.g. Melnik and Sparks, 1999; Lensky et al., 2004; Nakanishi and Koyaguchi, 2008;
Kozono and Koyaguchi, 2012) and upper conduit geometric configurations characterized by
multiple pathways (e.g. Lavallée et al., 2012; Reubi et al., 2015).
Two magma chambers located at different depths characterize the feeding system of Fuego de
Colima volcano (Fig. 1b), with roofs located at ca. 6 (shallow magma chamber) and ca. 15 km
(deep magma chamber) of depth, as indicated by petrographic studies (Macìas et al., 2017) and
geophysical data (Spica et al., 2017).
The purpose of this study is to investigate the existence of pattern of oscillations in discharge rates
during the 1998-2018 erupted activity at Fuego de Colima volcano. The available geological,
geophysical, and petrological data for this recent activity provide a remarkable opportunity to
improve the characterization and our understanding about the physical processes underlying cyclic
extrusion of lava domes. In particular, we used thermal remote sensing data along with published
effusion rates for reconstructing the oscillatory magma discharge rate behaviour of effusive activity
at Colima.
The availability of satellite thermal images in the last decade has strengthened the use of thermal
data for observing volcanic activity (e.g. Ramsey and Harris, 2012), especially in studying the
relationships with lava discharge rates (Coppola et al., 2009; Harris et al., 2010; Garel et al., 2012).
Coppola et al. (2013) propose that the radiant density of effusive/extrusive activity can be used to
estimate lava discharge rates and erupted volumes by means of empirical relationship based on $SiO_2$
content of the erupted lava. Although it is still under debate, the so-called "thermal approach"



(Dragoni and Tallarico, 2009) offers a good way for monitoring volcanic activity, especially when
direct observations are limited or absent.
Here we focus our attention to dynamics of fluctuations in magma discharge rate on various
timescales at Fuego de Colima volcano during 1998-2018. By using time series analytical
techniques (i.e. Fourier and wavelet analysis) we have identified three fundamental periodicities in
subsets of the time series: i) long-term (ca. 1.5-2.5 years), ii) intermediate-term (ca. 5-10 months),
iii) short-term (ca. 2.5-5 weeks), similar to those observed at many lava-dome eruptions (e.g. Costa
et al., 2012; Melnik and Costa, 2014; Christopher et al., 2015). These periodicities were compared
with numerical simulations provided by the model of Melnik and Sparks (2005) as generalized by
Costa et al. (2007a) for accounting the presence of a shallow dyke, and Melnik and Costa (2014) for
describing the control of a coupled dual chamber system. Numerical modelling of the different parts
of the pumbling system successfully reproduced the first-order cyclic behaviour of Fuego de
Colima during the 1998-2018 erupted activity. Our results highlighted that the dual magma
chamber dynamics controls the long-term periodicity evident during 2002-2006 and 2013-2016,
while the single magma chamber dynamics are more effective to explain the intermediate-term
periodicity in the same periods. Finally, the shallow dyke dynamics regulate the multi-week cycles
observed during 2002-2006 and 2011-2016.
The present work is divided in five main sections. The first describes the historical activity of the
Fuego de Colima, with particular attention to the recent period, from 1998 to 2018. The second
section describes the methods applied to the dataset composed of the satellite thermal data
integrated with published data. In particular, the Fourier analysis (including the discussion of its
limitations), the wavelet analysis with the definition of the wavelet transform, the choice of a
wavelet mother function, and the edge effects due to finite-length time series. It also includes the
use of the Melnik and Sparks (2014) model. The third section is dedicated to the input and target
data used for numerical simulations. The fourth presents the results obtained by the spectral and
wavelet analyses. This latter allows to establish significance levels for the wavelet power



spectrum. The periodicities observed in this spectrum were compared to the results obtained by
numerical simulations. The last fifth section contains a discussion on the eruptive behaviour
occurred at Fuego de Colima during 1998-2018, providing new insights from the observed data and
non-linear models.

**2. The historical activity of Fuego de Colima volcano**

Since historical times Fuego de Colima represents the most active volcano in Mexico, posing a
serious threat to all surrounding populations (Cortés et al., 2005; Gavilanes-Ruiz et al., 2009;
Bonasia et al., 2011). The earliest accounts of the volcano activity can be found in Historia Antigua
de Mexico (Clavijero, 1780), where the destructive effects of its explosive activity are carefully
described (Bretón-Gonzales et al., 2002). The historical activity of Fuego de Colima was described
and interpreted by several authors (Luhr and Carmichael, 1980; Medina-Martínez, 1983; De la
Cruz-Reyna, 1993; Bretón-Gonzales et al., 2002; Luhr, 2002). The Fuego de Colima has shown a
transitional eruptive behaviour spanning from effusive to explosive activity, dominated by dome
growth and Vulcanian eruptions. Occasionally sub-Plinian events occurred (1576, 1606, 1690, 1818
and 1913), indicating a recurrence time of approximately 100 years (De la Cruz-Reyna, 1993; Luhr,
2002; Saucedo et al., 2005; Gavilanes-Ruiz et al., 2009; Massaro et al. 2018a). The sub-Plinian
event occurred in 1913 (Saucedo et al., 2010) is the largest historical eruption and it has been used
as benchmark for volcanic hazard studies (Martin Del Pozzo et al., 1995; Saucedo et al., 2005;
Bonasia et al., 2011).

2.1. The 1998-2018 eruptive activity
The 1998-2018 is the only period of post 1913 activity for which there is sufficiently available
information to explore the cyclic activity of Fuego de Colima. Different periods of effusion (domes
and lava flows) punctuated by Vulcanian eruptions and dome collapses characterised the volcano



activity between 1998 and 2018 (Savov et al., 2008; Varley et al., 2010a; Hutchinson et al., 2013;
Mueller et al., 2013; Zobin et al., 2015; GVP, 2017). The duration of extrusive activity and magma
discharge rate varied through time, that was generally divided into five eruptive phases up to 2015;
I) 1998-1999; II) 2001-2003; III) 2004-2005; IV) 2007-2011; V) 2013-2015 (Zobin et al., 2015;
Aràmbula-Mendoza et al., 2018).
The first dome extrusion started in November 1998, and quickly filled the 1994 explosion crater,
forming lava flows that descended the southern flanks of the Fuego de Colima cone during most of
1999 (> 5 (m$^3$ s$^{-1}$) in average for Mueller et al., 2013; 4.11 (m$^3$ s$^{-1}$) in average for Reubi et al.,

148    2013).

At the beginning, this dome grew rapidly (ca. 4.4 m$^3$ s$^{-1}$) reaching a volume of ca. $3.8 \times 10^5$ m$^3$ in
24 hours. During this period the effusion rate reached a peak value around 30 (m$^3$ s$^{-1}$) (Navarro-
Ochoa et al., 2002; Zobin et al., 2005; Reubi et al., 2015) and showed a cyclic damped behaviour
soon after. During 1999-2001 a series of explosions destroyed the dome and excavated a large
apical crater (Bretòn-Gonzales et al., 2002).
A slow outpouring of lava (< 1 (m$^3$ s$^{-1}$) for Mueller et al., 2013; 0.17 (m$^3$ s$^{-1}$) for Reubi et al., 2013;
2015) resumed in May 2001 and continued for 22 months. In February 2002, the lava dome
overflowed the crater rims producing lava flows. During this eruptive phase, the magma extruded
from three separate vents with only minor explosive activity, at a rate of ca. 0.9 (m$^3$ s$^{-1}$) (GVP,
2002). Vulcanian explosions dismantled the dome during July and August 2003 (GVP, 2003).
In September 2004, low-frequency seismic swarms heralded the onset of the new effusive phase
(Varley et al., 2010a; Arámbula-Mendoza et al., 2011; Lavallée et al., 2012) with a small increase
in average discharge rate of 0.6 (m$^3$ s$^{-1}$) (Reubi et al., 2013; 2015). The lava dome building occurred
from the end of September until the beginning of November, with a magma effusion rate up to 7.5
(m$^3$ s$^{-1}$) in October (Zobin et al., 2008; 2015). The effusive activity was accompanied and followed
by intermittent Vulcanian explosions. The explosive activity diminished in intensity during
December 2004-January 2005. From February to September 2005, effusion and large explosions



occurred.
In the following months, small, short-lived domes were observed, with an estimated effusion rate
between 1.2 – 4.6 (m$^3$ s$^{-1}$) (Varley et al., 2010b; Reubi et al., 2015). In May and June, the explosive
activity produced pyroclastic density currents reaching distances up to 5.4 km from the volcano
summit (Varley et al., 2010a).
In February 2007, a new lava dome began to grow and explosions were reported in the period
between January 2009 and March 2011. The 2007-2011 period of dome extrusion represents the
slowest growth rate in the recent history of Fuego de Colima. Hutchinson et al. (2013) calculated a
mean effusion rate of ca. 0.02 (m$^3$ s$^{-1}$) from 2007 to 2010 using digital photographic data, in good
accordance with Zobin et al. (2015) that reported extrusion rates of 0.03 (m$^3$ s$^{-1}$) during 2007.
Mueller et al. (2013) estimated the magma extrusion rate between 0.008 ± 0.003 (m$^3$ s$^{-1}$) to 0.02 ±
0.007 (m$^3$ s$^{-1}$) during 2010, which dropped down to 0.008 ± 0.003 (m$^3$ s$^{-1}$) again in March 2011. On
21 June 2011 an explosion heralded the cessation of dome growth and marked the end of the
effusive period.
After 1.5 years of rest, in January 2013 a sequence of explosions cored out the 2011 dome and
generated pyroclastic density currents that reached distances of up to 2.8 km from the summit
(GVP, 2013). From March to October, the calculated discharge rate was in the range of 0.1 – 0.2
(m$^3$ s$^{-1}$) (Reyes-Dàvila et al., 2016). Successively, the mid-low explosive activity took place up to
February-March 2014, until a new pulse of magma observed in July, with an approximate rate of 1-
2 (m$^3$ s$^{-1}$) (Aràmbula-Mendoza et al., 2018). On January 11, 2015, a new lava dome was observed
inside the crater (Thiele et al., 2013) and its growth continued until July, with effusion rate of ca.
0.27 (m$^3$ s$^{-1}$) (Zobin et al., 2015). Between 10-11 July 2015 the recent dome was destroyed by the
most intense activity since the 1913 eruption (Capra et al., 2016; Reyes-Dávila et al., 2016). In the
2013-2015 period, the average extrusion rate was of ca. 0.2 (m$^3$ s$^{-1}$) (Thiele et al., 2017), with peak
values > 10 (m$^3$ s$^{-1}$) (Varley, 2015). After that, the eruptive activity ceased until January 2016 when
daily ash plumes started to occur along with active lava flows and explosions. In early July a new

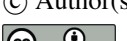



dome began to grow, overtopping the crater rim. A large explosion was recorded on 10 July 2016,
followed by daily and multiple-daily ash plumes up to the end of year. Multiple flows descended
from lava dome during September-December. In 2017 frequent strong explosions and ash emissions
were recorded until March. Through June decreasing seismicity and minor landslides were reported
with no evidence of effusive activity or new dome growth (GVP, 2017). Here we provide a more
systematic overview of the 1998-2018 erupted activity, obtained by satellite thermal data along with
some published data, explained in the following section.

**3.  Methods**
We analysed the thermal energy spectrum of Fuego de Colima volcano available from March 2000
to October 2018, detected Middle Infrared Observation of Volcanic activity (MIROVA) hot-spot
detection system (Coppola et al., 2016). The period 1998-1999 was integrated using published
discharge rates (Navarro-Ochoa et al., 2002; Zobin et al., 2005). The MIROVA NRT system is
based on the near real time (NRT) analysis of the MODerate resolution Imaging Spectroradiometer
(MODIS) data, distributed by the LANCE-MODIS data system (http://modis.gsfc.nasa.gov/).
The thermal emission from an object is attenuated by the atmosphere resulting from absorption by
gases and scattering by particles. MIROVA system focuses on the Middle InfraRed region (MIR),
which shows the lowest attenuation levels, to better detect and analyse thermal radiation emitted
from volcanic sources. While the standard MODIS forward processing delivers Aqua and Terra
images within 7-8 hours of real time, LANCE-MODIS allows for the creation of MIROVA radiant
flux timeseries within 1-4 hours from the satellite overpass (www.mirovaweb.it). This thermal data
collection was converted into lava discharge rate estimates and integrated with some published data
in order to reconstruct the weekly mean discharge rate spectrum from 1998 to 2018 (Fig. 2a).
In this work, we refer to Coppola et al. (2013), who describes the relationship between the heat lost
by lava thermal radiance variations and discharge rates, by means of a unique, empirical parameter.





They compared the energy radiated during several distinct eruptions to the erupted lava volumes
($m^3$). The relationship between the Volcanic Radiated Energy (*VRE*) and the erupted volume was
defined by introducing the concept of radiant density ($c_{rad}$, in J $m^{-3}$). This parameter is analysed as a
function of the $SiO_2$ content and the bulk rheological properties of the related lava bodies. It is
strongly controlled by the characteristic thickness of the active lavas at the time of a satellite
overpass, whereas the effects of variable degree of insulation, morphology and topographic
conditions produce only a limited range of variability (±50%) (Coppola et al., 2013). For the Fuego
de Colima we used a value of $c_{rad}$ = 3.90 × $10^7$ (J $m^{-3}$) with a $SiO_2$ content of 59.6% (Savov et al.,
2008; Coppola et al., 2013). We obtained the cumulative volumes of effusion per year (from 2000
to 2018) considering the ratio between the average *VRE* estimations and $c_{rad}$. It is important to stress
that the instrumental limit of the MIROVA system is not able to detect thermal anomalies below
0.5–1 MW. Since we used a radiant density ($c_{rad}$) of 3.90 × $10^7$ (J $m^{-3}$), the minimum reliable value
of discharge rate is 0.01 ($m^3$ $s^{-1}$) (Coppola et al., 2013). As reported by Coppola et al. (2016), the
thermal data obtained from MIROVA are not correct due to the presence/attenuation of clouds. For
this reason, the estimates of effusion rates and volumes are to be considered as minimum estimates.
Because the 2002-2006 and 2013-2016 intervals are the most active in the analysed period, we
firstly applied the Fourier analysis to the monthly average of discharge rates (Fig. 2b) of these time
intervals, in order to explore the modal spectrum of the signal. Although Fourier analysis is well
suited to the quantification of constant periodic components in a time series, it cannot recognise
signals with time-variant frequency content. Whereas a Fourier Transform analysis may determine
all the spectral components embedded in a signal, it does not provide any information about timing
of occurrence. To overcome this problem, several solutions have been developed in the past
decades that are able to represent a signal in the time and frequency domain at the same time.
The aim of these approaches is to expand a signal into different waveforms with local time–
frequency properties well adapted to the signal structure (Cazellas et al., 2008). In order to get



information on the amplitude of the periodic signals within the Fuego de Colima (MIROVA) time
series, we performed a wavelet analysis by decomposing the weekly time series (Fig. 2a) into
time/frequency space (Fig. 3).
Wavelet analysis is a powerful tool largely used in many scientific fields (i.e., ecology, biology,
climatology, geophysics) and engineering. It is especially relevant to the analysis of non-stationary
systems (i.e., systems with short-lived transient components, Cazellas et al., 2008). For this study,
practical details in applying wavelet analysis were taken from Torrence and Compo (1998) and
Odbert and Wadge (2009).  It is worth noting that wavelet analysis considers a wave that decays
over a finite time and whose integral over infinite time is zero. Many forms of wavelet (called
"wavelet functions" $\psi(\eta)$, or "mother functions", which depend on a non-dimensional time
parameter "$\eta$") have been designed for analytical use (Farge, 1992; Weng and Lau, 1994;
Daubechies, 1994), each with its own characteristics that make it suitable for certain applications.
The choice of the wavelet can influence the time and the scale resolution of the signal
decomposition.  Wavelet analysis is popular in geosciences (Trauth, 2006), as it does not require
any a priori understanding of the system generating the time series.
Our time series (weakly average discharge rates acquired mainly by the MIROVA system; Fig. 2a),
called $(x_n)$, has equal time spacing ($\delta t$ = 7 days) and number of points $n = 0…N-1$. Using the
approximately orthohogonal Morlet function as wavelet function $\psi(\eta)$ (it must have zero mean and
be localized in both time and frequency space; Farge, 1992), we here define the wavelet transform
$W_n(s)$ as the convolution of $x_n$ with a scale ($s$) and translated version of $\psi_0(\eta)$ (mother function). In
formula:
$$W_n(s) = \sum_{n'=0}^{N-1} x_{n'} \psi * \left[ \frac{(n'-n)\delta t}{s} \right]$$
    (1)

where the (*) indicates the complex conjugate. The scale $s$ should be equal to approximately $2\delta t$,





according to the Nyquist theorem. Therefore, the smallest wavelet we could possibly resolve is *2δt*,
thus we choose $s_0$ = 14 days. Generally, $\psi(\eta)$ is a complex function, therefore the wavelet transform
is also complex. It is possible to reconstruct the "local" wavelet power spectrum as the absolute-
value squared of the wavelet coefficients, $|Wn(s)|^2$. The way to compute the wavelet transform for a
time series is to find the Fourier transform of both the wavelet function (Morlet in our case) and the
time series. Following Torrence and Compo (1998), we made the normalization by dividing by the
square-root of the total wavelet variance ($\sigma^2$).
Usually, a periodic component in a time series may be identified in a power spectrum if it has
distinctly greater power than a mean background level (that would correspond to a Gaussian
background noise) (Odbert and Wadge, 2009). However, the spectra generated from many
geophysical systems indicate that the noise in time series data tends not to have a Gaussian
distribution (Vila et al., 2006) but it can be better described by coloured noise, specifically red noise
(Fougere, 1985). For this reason we use a simple model for red noise given by the unvariate lag-1
autoregressive or Markov process (Torrence and Compo, 1998) in order to determine the
significance levels for our wavelet spectrum.  These background spectra are used to establish a null
hypothesis for the significance of a peak in the wavelet power spectrum. The null hypothesis is
defined for the wavelet power spectrum considering that the time series has a mean power
spectrum: if a peak in the wavelet power spectrum is significantly above this background spectrum,
then it can be assumed to be a true feature with a certain percentage of confidence. For definitions,
"significant at the 5% level" is equivalent to "the 95% confidence level" (Torrence and Compo,
1998). The confidence interval is defined as the probability that the true wavelet power at a certain
time and scale lies within a certain interval about the estimated wavelet power (Torrence and
Compo, 1998). Because we deal with finite-length time series, errors occur at the beginning and end
of the wavelet power spectrum. A solution is to pad the end of the time series with zeroes to bring
the total length *N* up to the next-higher power of two, thus limiting the edge effects. However,



padding with zeroes introduces discontinuities at the endpoints and, especially towards larger
scales, decreasing the amplitude near the edges as more zeroes enter the analysis (Torrence and
Compo, 1998). The cone of influence (COI) is the region of the wavelet spectrum in which edge
effects become important. The criterion for applying wavelet analysis is very similar to those
employed with classic spectral methods. In other words, the wavelet transform can be regarded as a
generalization of the Fourier transform, and by analogy with spectral approaches, we compute the
local wavelet power spectrum as described above. Successively, this can be compared with the
"global" wavelet power spectrum which is defined as the averaged variance contained in all wavelet
coefficients of the same frequency  (Torrence and Compo, 1998; Cazellas et al., 2008).
Numerical simulations have been carried out using the magma flow model of Melnik and Costa
(2014), who generalized the model proposed by Melnik and Sparks (2005) for a magma chamber
connected to a dyke that evolves into a cylindrical conduit near surface. In particular, the model of
Melnik and Costa (2014) accounts for the possibility of a dual magma chamber system. The model
accounts for rheological changes due to volatile loss and temperature driven crystallization. These
processes are both effective during dome extrusion eruptions because of the typical low magma
ascent velocities (from millimetres to few centimetres per second), which can result in magma
transit times from days to weeks. These ascent times are often comparable with those of crystal
nucleation and growth, allowing efficient heat exchange between magma and wall rocks (Melnik
and Sparks, 1999; 2005; Costa et al., 2007c).

**4. Input and target data for numerical simulations**
4.1 Geometrical configurations of the magma plumbing system

The physical framework used in the Melnik and Costa (2014) the model (Fig. 1b) consists of two
magma chambers located at different depths, with chamber pressures $P_{chs}$ and $P_{chd}$ able to drive the



magma ascent in an elliptical cross-section volcanic conduit (approximating a dyke). Near surface
the conduit evolves into a cylinder at depth $L_T$ (named "transition level").
Numerical simulations were carried out considering the shallower magma chamber (single magma
chamber configuration) or the double magma chamber. The single magma chamber model
considers a conduit feeding system composed of a shallow dyke ($d_s$) that connects the magma
chamber to a shallower cylinder, in agreement with geological and geophysical evidence from
different volcanoes (Melnik and Sparks, 2005; Costa et al., 2007a; Melnik et al., 2008; Melnik and
Costa, 2014). The double magma chamber model includes the addition of a deep reservoir
connected to the shallow chamber through an elastic deep dyke ($d_d$) (Fig. 1b).
In order to reproduce the observed fluctuations in discharge rates recorded in some periods of the
1998-2018 erupted activity, we considered a discharge rate regime where the period of pulsations is
controlled by the elasticity of the shallow dyke, and a discharge rate regime where the periodicity is
controlled by the volume of the single or dual magma chamber(s) (Barmin et al., 2002; Melnik and
Sparks, 2005; Costa et al., 2007a; Melnik and Costa, 2014).
In Appendices A2 and A3 we reported some test simulations in order to show the control of the
most sensitive parameters (i.e. water content in magma, dyke dimensions, volume of magma
chamber, magma influx rate into the magma chamber) affecting the model outputs in case of the
single magma chamber model. The volumes of the magma chamber ($V_{ch}$) range from 20 to 50 km$^3$
and the width of the feeder dyke $2a$ from 200 to 400 m (Massaro et al., 2018a).
In Appendix A4 is shown the sensitivity test aimed to explore a broad range of chamber volumes
and aspect ratios in the case of double magma chamber configuration. The deep chamber has its top
at 15 km of depth, it is pressurised and fed from below by a constant influx $Q_{in,d}$. The volumes of
shallow magma chamber ($V_{chs}$) range from 30 to 50 km$^3$, and from 550 to 750 km$^3$ for the deep
magma chamber ($V_{chd}$), according to geophysical data (Cabrera-Espindola, 2010; Spica et al.,
2017). The aspect ratios for shallow and deep magma chambers ($AR_s$ - $AR_d$) varied from 1 to 2. For
each run included in the sections 1-3 of A4, we used a fixed influx $Q_{in,d} = 2.3$ (m$^3$ s$^{-1}$), and variable



widths of the deeper dyke ($2a_{0d}$) from 200 to 3000 m. The lower dyke thickness $2b_{0d}$ is not an input
data of the model as it changes as function of local pressure conditions, therefore it does not appear
in the diagrams. In Section 4 of A4 we show two sets of runs having $Q_{in,d}$ equal to 1 and 3 ($m^3 s^{-1}$)
respectively, and the following fixed parameters: $AR_s$ and $AR_d = 1$, $V_{chd} = 650$ km$^3$, $V_{chs} = 40$ km$^3$.

4.2 Petrological data

Erupted products at Fuego de Colima are chemically intermediate and primarily andesitic lavas with
ca. 61 wt.% $SiO_2$, (Lavallè et al., 2012). The observed dome growth phases are usually fed by
prolonged magma ascent times, which allow efficient degassing and crystallization. This is in
agreement with the low mean porosity (14-16% e.g Lavallè et al., 2012; Farquharson et al., 2015)
and low water contents of the products of the recent activity (2 wt. % for 1998-1999, Mora et al.,
2002; 0.1-2.5 wt. % for 1998-2005 products, Reubi and Blundy, 2008). Dome lava currently
erupted exhibits a range of crystallinities (phenocrysts, 20–30 vol.%; microlites, 25–50 vol.%), and
the groundmass constitutes as much as 68 vol.% (Luhr, 2002). The andesites show a porphyritic
texture with plagioclase (13–25 vol.%), orthopyroxene (2– 4 vol.%), clinopyroxene (3–4 vol.%) and
minor hornblende (<0.5%) and Fe–Ti oxides ( ca. 2 vol.%). Olivine occurs rarely as xenocrysts
(Lavallè et al., 2012). According to Costa et al (2007a), magma viscosity increases due to the
crystal fraction $\beta$ as described through the function $\Theta(\beta)$ (Costa et al., 2009):
$$\theta = \frac{1 + \left(\frac{\beta}{\beta*}\right)^{\delta}}{\left(1 - erf\left\{\frac{\sqrt{\pi}}{2\varepsilon}\frac{\beta}{\beta*}\left[1 + \left(\frac{\beta}{\beta*}\right)^{\gamma}\right]\right\}\right)^{2.5\beta*}}$$
(2)

where $\beta*$ represents the critical transition fraction, $\gamma$ is a measure of the steepness of the rheological
transition, $\varepsilon$ ($0 < \varepsilon < 1$) determines the value of $\Theta(\beta*)$. In principle $\beta, \gamma, \delta$ and $\varepsilon$ can be described a
function of the strain rate and crystal shape but here are assumed to be constant (Costa et al.,
2007a). As crystallization proceeds the remaining melt becomes more silica rich and viscosity




increases (Costa et al., 2007a). Table 1 summarises the value ranges used for the input parameters
of the model.

**5. Results**
In Figure 2 we showed the averages of discharge rates at Fuego de Colima volcano from November
1999 to October 2018. Here we define as "high" discharge rates values > 0.1 ($m^3 s^{-1}$) (highlighted as
dark blue areas). All values below > 0.1 ($m^3 s^{-1}$) are considered "low" discharge rates (light blue
areas). Volcanological observations are reported at the top and the bottom of the diagram. In
addition, we distinguished between lava flows and lava domes accordingly to the dominant
emplacement style typical of each eruption, and between "low" (i.e. ash plumes, gas emissions) and
"high" (i.e. strong explosions, Vulcanian eruptions) magnitude explosive activity.
The weekly average of discharge rates represents the complete dataset used in this study, and is
reported in Figure 2a. These data have been calculated by using the MIROVA data (black dots) for
the 2000-2018 period, and complemented with published data (blue crosses) for the 1998-1999
period (Navarro-Ochoa et al., 2002; Zobin et al., 2005). Even if the data detection of satellite
thermal energy represents a continuous spectrum of information, it is worth noting that it suffers of
some limitations connected to cloud covering, magma composition, rheology, and emplacement of
the investigated lava body due to topographic conditions (Harris and Rowland, 2009; Harris et al.,
2010; Coppola et al., 2013).
Figure 2b shows the monthly discharge rate spectrum from 1998 to 2018 using the MIROVA
dataset (black dots), integrated with available published data (blue crosses) (Navarro-Ochoa et al.,
2002; Zobin et al., 2005; Capra et al., 2010; Varley et al., 2010a; Sulpizio et al., 2010; James and
Varley, 2012; Hutchinson et al., 2013; Reubi et al., 2013; Varley, 2015; Reyes-Dávila et al., 2016;
Thiele et al., 2017; GVP, 2000; 2017). Figure 2c summarizes the yearly average of discharge rates





from MIROVA dataset, highlighting the good agreement with the available average estimation of
yearly discharge rates from literature (Mueller et al., 2013; Reyes-Dàvila et al., 2016; Aràmbula et
al., 2018; GVP, 1998-2017).

5.1 Fourier analysis
The Fourier analysis applied to 2002-2006 period showed two periodic components, $T_0 = 24.70$ and
$T_1 = 6.17$ corresponding to ca. 2 years and ca. 6 months, respectively (Appendix A1, Fig. a). For
2013-2016 we obtained similar results: $T_0 = 24.94$ and $T_1 = 6.23$ corresponding to ca. 2.1 years and
ca. 6 months, respectively (Appendix A1, Fig. b).

5.2 Morlet wavelet analysis
The wavelet analysis is well suited for investigations of the temporal evolution of aperiodic and
transient signals. Indeed, wavelet analysis is the time–frequency decomposition with the optimal
trade-off between time and frequency resolution (Lau and Weng, 1995; Mallat, 1998). The whole
analysed dataset is composed of 825 data points, representing the time evolution of the oscillating
components of the 1998-2018 eruptive activity (Fig. 2a). Figure 3a shows the normalised local
wavelet power spectrum of the signal. The colours scale for power values vary from light orange
(low values) to dark red (high values). The thick black contours represent the 95% confidence level.
The blue line indicates the cone of influence (COI) that delimits the region not influenced by edge
effects. From this analysis, it is easy to observe three main periodicities during 2002-2006 and
2013-2016 periods: i) long-term periodicity of ca. 1.5–2.5 years; ii) intermediate-term periodicity of
ca. 5-10 months; and, iii) short-term periodicity of ca. 2.5-5 weeks. The short-term periodicity is
also present in 2011 (Fig. 3a). Figure 3b shows the global wavelet spectrum corresponding to the
local wavelet power spectrum plotted in Fig. 3a. The green dashed line shows the position of the
best-fitting red noise model at the 95% confidence level.





### 5.3 Numerical simulations

Appendices A2-A4 provide some sensitivity tests in order to explore the effects of different parameters on discharge rate fluctuations for the single (A2-A3) and dual magma chamber model (A4). In particular, in Appendix A2 is reported the general steady-state solution of the numerical model, with both stable and unstable branches (e.g. Melnik et al., 2008; Nakanishi and Koyaguchi, 2008), showing that the cyclic behaviour can occur only between 2 and 4 ($m^3$ $s^{-1}$), for the fixed input data (panel (a)). Varying the width of the shallow dyke *2a* (from 200 to 400 m) and water content in the melt phase, we observed how the unstable branch changes its shape. This implies different periods of possible oscillations in discharge rate (panels (b)-(c)).

Appendix A3 provides a set of simulations carried out varying the width of the shallow dyke *2a* (panel (a)). The resulting periodicities vary from ca. 1000 days (*2a* = 200 m) ca. 500 days (*2a* = 300 m) to ca. 250 days (*2a* = 400 m). These results highlight negative correlation between dyke widths and periods of oscillation (Costa et al., 2007a). In this case, the variable widths influence the intensity and periodicity of discharge rates: the wider the dyke, the lower the intensity and periodicity of discharge rates. Differences in the amplitude of oscillations are observed in panel (b), highlighting a positive correlation between the volume of the magma chamber $V_{ch}$ and periodicities. Periodicities of ca. 500 days correspond to 20 - 30 $km^3$, while larger values of ca. 970 and ca. 1176 days are provided for 40 and 50 $km^3$, respectively.

In panel (c), we reported also a set of simulations considering the modelled discharge rate controlled by the elasticity of the shallower dyke with fixed influx rates $Q_{in}$ (in the range of 0.01 - 0.1 $m^3$ $s^{-1}$). Although this set of runs showed damped oscillations, which do not represent the periodicities observed at Fuego de Colima, it shed lights on the model's output variability in relation to the boundary conditions.

Appendix A4 contains the sensitivity tests for the dual magma chamber model. As reported in Melnik and Costa (2014), the dual chamber model shows cyclic behaviour with a period that



depends on the intensity of the influx rate and the chamber connectivity (described as the horizontal
extent of the dyke connecting the two chambers). For a weak connectivity, the overpressure in the
deeper chamber remains nearly constant during the cycle and the influx of fresh magma into the
shallow chamber is also nearly constant. For a strong connectivity between the two chambers, their
overpressures increase or decrease during the cycle in a synchronous way. Influx into the shallow
chamber stays close to the extrusion rate at the surface (Melnik and Costa, 2014). We explored
different cases considering various fixed parameters as follow: *i*) volumes of the shallow and deep
magma chambers ($V_{chs}$ = 40 km$^3$, $V_{chd}$ = 650 km$^3$); *ii*) aspect ratios ($AR_s$ = 1, $AR_d$ = 1) and the deep
magma chamber volume ($V_{chd}$ = 650 km$^3$); *iii*) aspect ratios ($ARs$ = 1, $ARd$ = 1) and the shallow
magma chamber volume ($V_{chs}$ = 40 km$^3$). For *i*), *ii*) and *iii*) cases, the deep influx rate $Q_{in,d}$ has fixed
values from 3 to 1 m$^3$/s.  In conclusion, these sensitivity tests showed the passage from weakly
connected magma chambers (lack of simultaneous oscillation of $Q_{in,s}$ and $Q_{out}$) when $2a_{0d}$ = 200 m
to strongly connected magma chambers (synchronous oscillations of $Q_{in,s}$ and $Q_{out}$) when $2a_{0d}$ =
3000 m.
Figure 4a shows a representative example of time-dependent solution for a discharge rate controlled
by the elasticity of the shallower dyke. Simulations were carried out using fixed values of pressure
(blue line) and influx rate ($Q_{in,s}$ = 3.5 m$^3$/s) (green line) at the source region of the shallower dyke.
The dyke is long 5500 m, it has width $2a$ = 400 m and thickness $2b$ = 2 m and a dyke-cylinder
transition at 1300 m of depth. The magma chamber volume is fixed to 30 km$^3$. Solutions present
periodicities from 16 to 40 days in agreement with the weekly periodicities of ca. 38-18 days (ca.
2.5-5 weeks) derived from the wavelet analysis (Fig. 3a).
Figure 4b describes a representative example of the single magma chamber model simulations. We
set the magma feeding system composed of a dyke long 6500 m, having a width $2a$ = 600 m,
thickness $2b$ = 4 m, and a dyke-cylinder transition fixed at 1000 m of depth. The chamber has a
volume fixed to 30 km$^3$ and receives a constant $Q_{in,s}$ = 2.3 (m$^3$ s$^{-1}$). The transient solution is
accounted for the discharge rate controlled by the magma chamber volume, showing an



intermediate-term periodicity of ca. 220 days, in agreement with the intermediate-term periodicity
of ca. 146-292 days (ca. 5-10 months) obtained from the wavelet analysis (Fig. 3a).
Figure 4c reports a representative example of the solution obtained with the dual magma chamber
model in order to assess the effect of the deep chamber on the discharge rate. We fixed the volumes
of deep and shallow magma chamber at 40 and 650 km$^3$, respectively. The shallow dyke is 6500 m
long with a width $2a = 260$ m and thickness $2b = 4$ m. The deep dyke has a width $2a_{0d} = 500$ m, and
a deep influx rate $Q_{in,d} = 2.3$ (m$^3$ s$^{-1}$). A cyclic behaviour of ca. 825 days is observed, reaching a
peak discharge rate of ca. 6 (m$^3$ s$^{-1}$). This result is in agreement with the long-term periodicity of ca.
547-913 days (ca. 1.5 - 2.5 years) derived from the wavelet analysis (Fig. 3a).

**6.   Discussions**
In recent years, many studies have focused on magma flow dynamics in volcanic conduits during
lava dome building eruptions (Melnik and Sparks, 1999; Wylie et al., 1999; Barmin et al., 2002;
Melnik and Sparks, 2002; 2005; Costa et al., 2007a,b; Nakanishi and Koyaguchi, 2008; Kozono and
Koyaguchi, 2012), highlighting periodic variations in discharge rate due to the transition from low
regime (allowing efficient crystals grow leading to increase in magma viscosity) to high regime
(with negligible crystallization). This difference in discharge rates can be of orders of magnitude,
with strongly non-linear responses to the variation of governing parameters from the volcanic
system. This behaviour allows periodic oscillations of the discharge rate (Nakada et al., 1999; Watts
et al., 2002), as observed in different dome extrusion eruptions (e.g. Mt St. Helens, Santiaguito,
Montserrat, (Melnik et al., 2008). Although each volcano usually shows its complex pattern of
discharge fluctuations, the cause can be explained as the superimposition of long, intermediate, and
short-term effects of the coupled magma chamber(s) and conduit dynamics. The long-term
oscillations in discharge rate are function of magma chamber size, magma compressibility, and of
the amount and frequency of magma recharge and withdrawal (Barmin et al., 2002; Costa et al.,



2007b; Melnik et al., 2008; Costa et al., 2013; see Appendices A2-A3). The short-term and
intermediate oscillation dynamics can also superimpose to the main long-term periodicity, through
small changes in magma temperature, water content, and kinetic of crystallization during magma
transit in the conduit (e.g., Melnik et al., 2008). The aforementioned eruptive behaviour
characterized also the Fuego de Colima activity in the 1998-2018 period, as demonstrated by the
wavelet analysis of satellite thermal data. It is worth noting that the oscillating behaviour is not
regular, having a period, between 2007 and 2012, that does not show any significant periodicity
(Fig. 3a). During this period the volcano enter in an almost quiescent status with very low discharge
rates. This period of low discharge rates is punctuated by low explosive activity, triggered by dome
collapse or pressurization of the upper conduit.
In order to investigate the relationship between the periodic components observed in wavelet
analysis and the dynamics of the Fuego de Colima feeding system, we run simulations using the
numerical model Melnik and Costa (2014) (Fig. 4). The model can reproduce the results of the
wavelet analysis in terms of observed periodicities, allows us to relate short-, intermediate- and
long-term oscillations in discharge rates to the dynamics of upper conduit, shallow magma
chamber, and coupled shallow and deep magma chambers, respectively. This implies that the
pressurization of the deep magma chamber has cascade effects on the whole feeding system of the
Fuego the Colima, similarly to what observed in other recent lava dome eruptions (i.e. Montserrat;
(Melnik and Costa, 2014). It is of particular interest that the best output with the dual magma
chamber model indicates that chambers do not oscillate simultaneously ("decoupled oscillation";
Fig. 4c). This accounts for the coexistence of long and intermediate periodicities in the 2002-2006
and 2013-2016, which would have not been possible in case of synchronous oscillation of the two
magma chambers. It means that the influx of magma from the deeper into the upper feeding system
induces pressurization to the shallow system (magma chamber + conduit), which starts to oscillate
following its own periodicities.
Although the presented data provide, for the first time, a framework able to explain the periodic



behaviour of effusive activity at Fuego de Colima volcano, both numerical model and wavelet
analysis suffer of some limitations that need to be taken into account in interpreting the results:

*i)* the available data of discharge rates and dome volumes collected for the 1998-2018

period do not have the same quality. For this reason, this lead us to extract only averages of

discharge rate for the entire period, with biasing effects to lower amplitudes;

*ii)* a common weakness of the spectral and wavelet analysis techniques is their inability to

distinguish the source of any given periodic component (i.e. whether it is a signal from a

volcanic process, an external process or if it is noise in the data). Elucidating the exact

mechanism requires competing robust models and multiple independent field observations

(Odbert and Wadge, 2009);

*iii)* assumptions behind the numerical model imply several limitations, such as those due to

the constant value of the dyke width and simplified Newtonian rheology. The first

assumption greatly oversimplifies the physics. In the case of large overpressures, stress at

the dyke tips will exceed the fracture toughness of the rocks and the dyke will expand

horizontally (Massaro et al., 2018b), reaching some equilibrium configuration. When the

deep chamber deflates, overpressure in the deeper dyke will decrease and, as flow rate

decreases, magma at the dyke tips can solidify, leading to a decrease in $2a_{0d}$ (Kavanagh and

Sparks, 2011; Melnik and Costa, 2014). In addition, a more realistic estimate of the magma

viscosity during lava dome eruptions should account for the coupling with energy loss,

viscous dissipation, and stick–slip effects (e.g. Costa and Macedonio, 2005; Costa et al.

2007c; 2013).


**7. Conclusions**
The coupling of wavelet analysis and numerical modelling allowed the deciphering of eruptive
behaviour of Fuego de Colima in the period 1998-2018, as revealed by satellite thermal data. Three
periodicities emerged from the study: i) long-term ii) intermediate-term, and, iii) short-term.



The long-term periodicity extracted from wavelet analysis is ca. 913-547 days (ca. 1.5-2.5 years),
which can be replicated by the dual magma chamber model that provided a periodicity of ca. 1000-
500 days. The intermediate-term periodicity obtained from wavelet analysis (ca. 146-292 days, 5-10
months) can be replicated by the single magma chamber model, which provides a periodicity of ca.
220 days. The short-term periodicity of ca. 18-38 days (ca. 2.5-5 weeks) is matched by model
outputs considering the dynamics of the upper conduit (ca. 16-40 days). The depicted behaviour of
effusive activity at Fuego de Colima is here presented for the first time, showing how the volcano
presents similarities with eruptive dynamics of other recent lava dome eruptions (i.e. Montserrat,
Costa et al., 2013).
**Code availability**
Melnik and Costa (2014) code is a research software and is not still available for distribution as it
lacks of documentation. It can be used by contacting the authors under their supervision.
**Data availability**
The original thermal dataset is available on www.mirovaweb.it. Excel worksheets can be obtained
by contacting the authors.
**Appendices**
**Appendix A1.** Results of the Fourier analysis. (a) The 2002-2006 period shows two main periodic
components, $T_0 = 24.70$ and $T_1 = 6.17$ months, corresponding to ca. 2 years and ca. 6 months,
respectively; (b) The 2013-2016 period shows similar results: $T_0 = 24.94$ and $T_1 = 6.23$ months,
corresponding to ca. 2.1 years and ca. 6 months, respectively.

**Appendix A2.** Sensitivity tests for steady state solutions of discharge rate vs chamber pressure (top)
and time evolution of discharge rates (bottom). These solutions are referred to the following main
input parameters: i) dyke thickness $2b = 40$ m as the conduit diameter at the top ($D=2b$), the
transition from the dyke to cylindrical conduit $L_T = 500$ m below the surface, the length of the dyke
$L_d = 6$ km, and the volume of the magma chamber $V_{ch} = 50$ km$^3$. (a) General solution showing the
transient regime where the periodicity can occur; (b) Solutions influenced by the dyke width $2a$
(from 200 to 400 m); (c) Solutions influenced by the proportion of the water content in the melt
($H_2O$ from 4 to 5 %).





**Appendix A3.** Sensitivity tests for transient solutions using the single magma chamber model. As a
reference these solutions have the same main input parameters used for A2. (a) Dependence of
discharge rate on time considering the influence of the dyke width $2a$ (from 200 to 400 m); (b)
Influence of the magma chamber volume $V_{ch}$ (from 20 to 50 km$^3$); (c) Dependence of discharge rate
on time considering the dyke elasticity. Each curve shows a solution with a constant influx rate $Q_{in}$
(in the range of 0.01- 0.1 m$^3$ s$^{-1}$).

**Appendix A4.** Sensitivity tests for transient solutions using the dual magma chamber model. The
shallow feeding system has dyke with a width $2a = 200$ m, $2b = 40$ m, and $L_T = 500$ m. The
cylindrical conduit diameter $D = 2b$. For each diagram, is indicated the outflow ($Q_{out}$; black red and
green lines), the flux entering into the shallower magma chamber ($Q_{ins}$; blue line) and periods in
days (T). Runs of Section 1-2-3 have $Q_{in,d} = 2.3$ (m$^3$ s$^{-1}$).

- *Section 1*) The volumes of the shallow and deep magma chambers are fixed to 40 km$^3$ and
650 km$^3$, respectively. A set of runs is carried out for three different aspect ratios ($AR$) of the
shallow and deep chambers ($ARs = 1$; $ARd = 1$, $ARs = 2$; $ARd = 1$, $ARs = 2$; $ARd = 1.5$)
considering three widths of the deeper dyke ($2a_{0d} = 200$ m - black line, 1000 m - red line,
3000 m - green line).

- *Section 2*) The volume of the deeper magma chamber and the aspect ratios of both shallow
and deep chambers are fixed to 650 km$^3$ and $ARs = ARd = 1$. A set of runs is provided for
three different shallow chamber volumes ($V_{chs} = 30$ km$^3$, 40 km$^3$, 50 km$^3$) considering three
widths of the deeper dyke ($2a_{0d} = 200$ m - black line, 1000 m - red line, 3000 m - green
line);

- *Section 3*) The shallow chamber volume and the aspect ratios of both shallow and deep
chambers are fixed to 40 km$^3$ and $ARs = ARd = 1$, respectively. A set of runs is carried out
for three deep chamber volumes ($V_{chd} = 550$ km$^3$, 650 km$^3$, 750 km$^3$) considering three
widths of the deeper dyke ($2a_{0d} = 200$ m - black line, 1000 m - red line, 3000 m - green
line).

- *Section 4*) The shallow and deep chamber volumes are fixed to 40 km$^3$ and 650 km$^3$,
respectively. Two set of runs are carried out for $Q_{in,d}$ equal to 1 and 3 (m$^3$ s$^{-1}$). The aspect
ratios ($AR$) of the shallow and deep chambers are both equal to 1, considering three widths
of the deeper dyke ($2a_{0d} = 200$ m - black line, 1000 m - red line, 3000 m - green line).




**Author's contribution**

SM and AC compiled the numerical simulations and formulated the adopted methodology. DC provided and processed the satellite thermal data. LC provided the volcanological data. SM and RS wrote the manuscript with the input of all the co-authors. All authors worked on the interpretation of the results.

**Competing interests**

The authors declare that they have no conflict of interest.

**Acknowledgments**

SM thanks Centro de Geociencias of Queretaro (UNAM, Mexico) for the hospitality during the period of research at Fuego de Colima volcano, the Doctoral Course in Geoscience of University of Bari (Italy) for the partial financial support and Dr. F. Loparco for the help with the Python coding. LC was supported by PAPIIT-UNAM n° 105116 project.

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





## Tables

Table 1: Input parameters used in numerical simulations.

| Notation | Description | Value |
|---|---|---|
| $c_o$ | Concentration of dissolved gas (wt.%) | 0.05-0.06 |
| $C_f$ | Solubility coefficient ($Pa^{-1/2}$) | $4.1 \times 10^{-6}$ |
| $C_m$ | Specific heat ($J\ kg^{-1}\ K^{-1}$) | $1.2 \times 10^{3}$ |
| $I_0$ | Max nucleation rate ($m^{-3}\ s^{-1}$) | $3 \times 10^{10}$ |
| $L_*$ | Latent heat of crystallization ($J\ kg^{-1}$) | $3.5 \times 10^{5}$ |
| $\mu_g$ | Gas viscosity (Pa s) | $1.5 \times 10^{-5}$ |
| $\rho_m$ | Density of the melt phase ($kg\ m^{-3}$) | 2300-2500 |
| $\rho_c$ | Density of the crystal ($kg\ m^{-3}$) | 2700-2800 |
| $T_{ch}$ | Temperature in the magma chamber (K) | 1150 |
| $P_{ch}$ | Pressure in the magma chamber (MPa) | 130 - 210 |
| $\beta_{ch*}$ | Crystal content in magma chamber | 0.35-0.45 |
| $\rho_r$ | Host rock density ($kg\ m^{-3}$) | 2600 |
| $G$ | Host rock rigidity (GPa) | 6 |
| $v$ | Poisson's ratio | 0.25 |
| $\varepsilon$ | | 8.6 |

*Conduit geometry parameters using a single magma chamber model*

| | | |
|---|---|---|
| $D$ | Diameter of the cylindrical conduit | 30-40 |
| $L_T$ | Dyke-cylinder transition depth (m) | 1300-500 |
| $2a$ | Width of the dyke (m) | 200 – 600 |
| $2b$ | Thickness of the dyke (m) | 4-40 |
| $L$ | Depth of the magma chamber (top) (m) | 5500-6500 |
| $V_{ch}$ | Chamber volume ($km^3$) | 20-50 |
| $AR$ | Aspect ratio of the magma chamber | 1-2 |
| $Qin,s$ | Influx into the shallow magma chamber ($m^3\ s^{-1}$) | 0.01-3.5 |

*Parameters used for simulations carried out with dual magma chamber model*

*Deep magma chamber*

| | | |
|---|---|---|
| $2a_{0d}$ | Width of the deeper dyke (m) | 200 – 3000 |
| $L_0$ | Depth of the deep magma chamber (top) (m) | 15000 |
| $ARd$ | Aspect ratio of the deep magma chamber | 1-2 |
| $V_{chd}$ | Deep chamber volume ($km^3$) | 550-750 |
| $\Delta P$ | Deep magma chamber overpressure (MPa) | 20 |
| $Qin,d$ | Influx into the shallow magma chamber ($m^3 s^{-1}$) | 1-3 |



**Figures Captions**

**Fig. 1.** (a) Digital elevation model of the Colima Volcanic Complex (NC = Nevado de Colima volcano; FC = Fuego de Colima volcano) and Colima Rift with the main tectonic and volcano-tectonic structures (modified from Norini et al. 2010). In the inset, the location of the Colima Volcanic Complex (CVC) within the Trans-Mexican Volcanic Belt (TMVB) is shown in the frame of the subduction-type geodynamic setting of Central America. (b) Schematic view of the conduit feeding system framework used for numerical simulations (modified after Melnik and Costa, 2014).

**Fig. 2.** Dataset about the averaged discharge rates of Fuego de Colima during 1998-2018, derived by the MIROVA thermal data (black points) and published data (blue crosses) (Navarro-Ochoa et al., 2002; Zobin et al., 2005; Reubi et al 2013; Mueller et al., 2013; Varley, 2015; Reyes-Dàvila et al., 2016; Theiele et al., 2017; GVP, 2002-2017). Values > 0.1 (m$^3$ s$^{-1}$) are considered to be as "high" (dark blue area) and values < 0.1 (m$^3$ s$^{-1}$) as "low" discharge rate (light blue area). The 0.01 (m$^3$ s$^{-1}$) is the threshold under which the MIROVA system does not provide reliable data (blue line); (a) Weekly average discharge rates. The boxes contain symbols of volcanological observations reported in literature; (b) Monthly average discharge rates; (c) Yearly average discharge rates.

**Fig. 3.** (a) Local wavelet power spectrum normalized by $1/\sigma^2$ ($\sigma^2$ in (m$^3$ s$^{-1}$)$^2$). The left axis is the period (in years). The bottom axis is time (in years). The shaded contours are at normalized variances of 0.5, 1, 2, and 4 (m$^3$ s$^{-1}$)$^2$. The black thick contour encloses regions of greater than 95% confidence for a red-noise process with a lag-1 coefficient of 0.72. It shows three orders of periodicities of: long-term (ca. 1.5-2.5 years), intermediate-term (ca. 5-10 months) during 2002-2006 and 2013-2016, and short-term (ca. 2.5-5 weeks) during 2001-2006 and 2011-2016. Blue line indicates the "cone of influence" where edge effects become important outside it; (b) Global wavelet power spectrum. The green dotted line represents the best-fitting red noise spectrum at the 95% confidence level.

**Fig. 4.** Results of numerical simulations. The physical framework of the conduit feeding system has deep and shallow chambers connected to surface via vertical elastic dykes evolving into non-elastic cylinder. The length of the shallow dyke $L_{ds}$ is in the range of 5500-6500 m. The passage to cylinder conduit $L_T$ occurs at ca. 1300-500 m below the cone. (a) Discharge rates vs. time considering the elasticity of the shallower dyke, with a width $2a = 400$ m and thickness $2b = 2$ m. The cylinder diameter $D = 30$ m. Two cases are shown: *i)* constant pressure (blue line) and *ii)* constant influx rate at the source region of the dyke, providing different periodicities of 16 and 40 days, in good agreement with the short-term (weekly) periodicities observed in Fig. 3a; (b) Discharge rate vs. time





using the single magma chamber model. The dyke width $2a$ = 600 and thickness $2b$ = 4 m. The
chamber has a volume $V_{ch}$ = 30 km$^3$, receiving a constant influx $Q_{in,s}$ = 2.3 (m$^3$ s$^{-1}$); Periodicity is of
ca. 220 days, in good agreement with the intermediate-term (monthly) periodicities observed in Fig.
3a; (c) Discharge rate vs. time using the dual magma chamber model. The aspect ratio of the
shallow and deep chambers ($ARs$ - $ARd$) are both equal to 1.3 and 1.4, respectively. The upper
feeding system has a chamber ($V_{chs}$ =30 km$^3$) connected to a dyke (width $2a$ = 260 m; $2b$ = 4 m)
evolving into a cylinder ($D$ = 30 m) at $L_T$ = 1000 m. The shallow chamber is connected to the deep
one ($V_{chd}$ = 500 km$^3$) through a feeder dyke ($2a_{0d}$ = 500 m). A constant $Q_{in,d}$ = 2.3 (m$^3$ s$^{-1}$) is
injected from below. Periodicity is in the range of ca. 825 days, in good agreement with the long-
term (yearly) periodicities observed in Fig. 3a.



















**Fig.1**

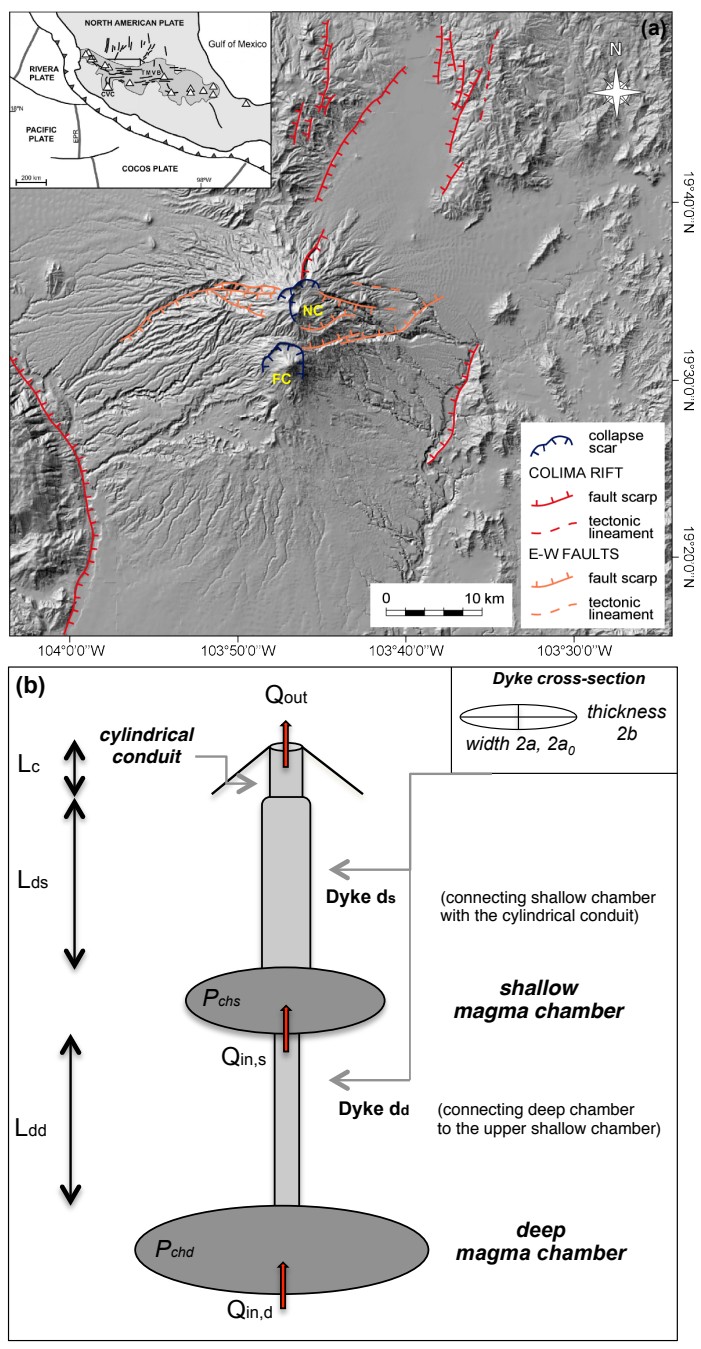






100    **Fig. 2**

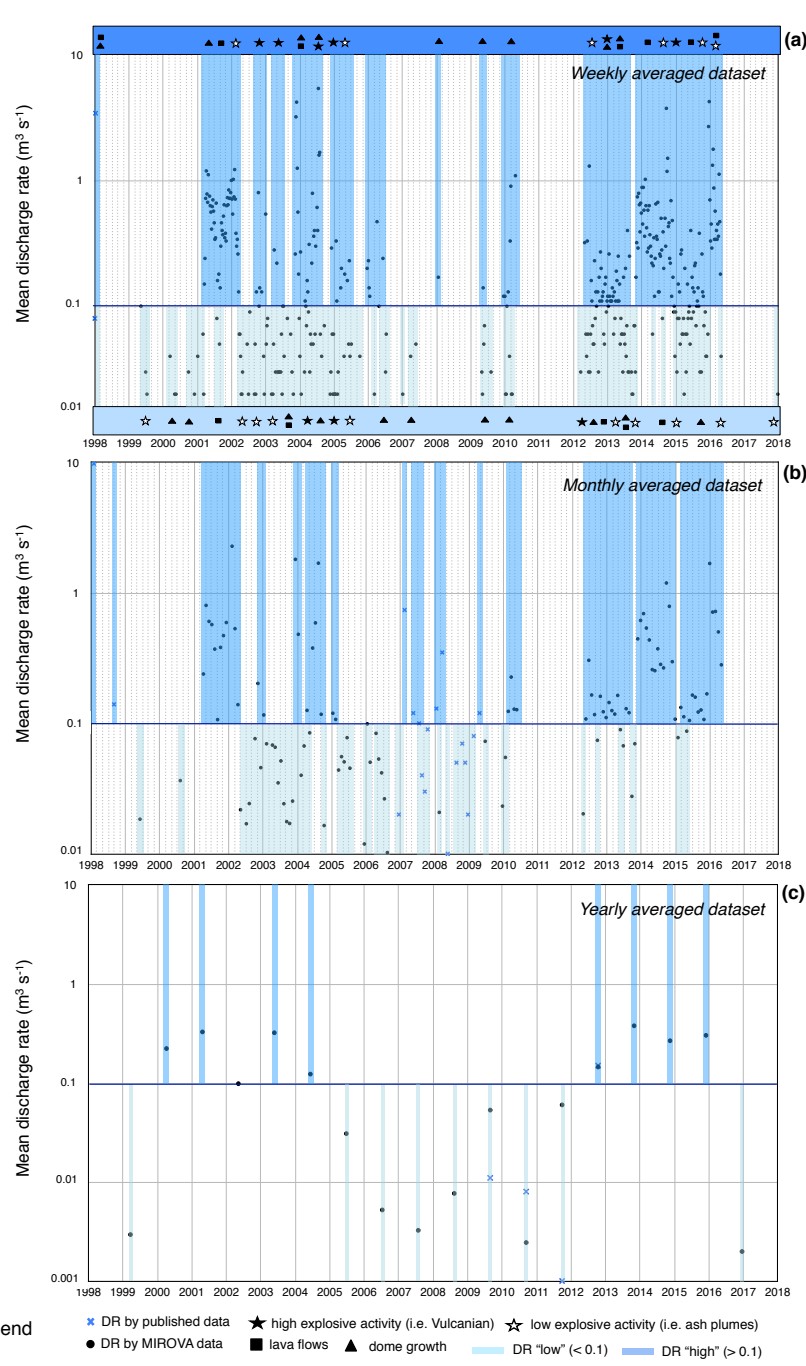

101

102





**Fig. 3**

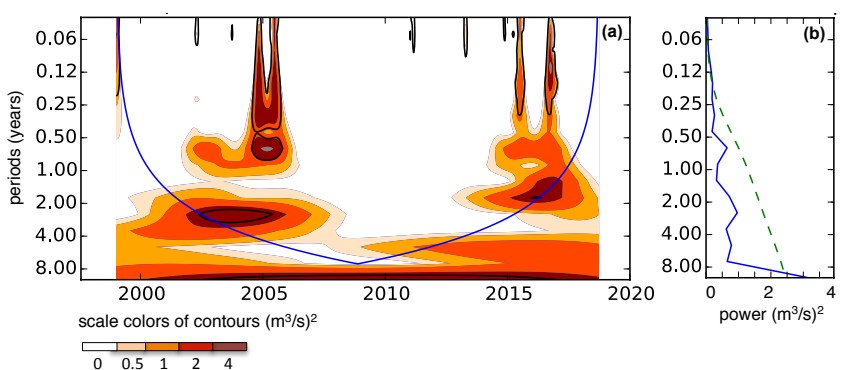













**Fig. 4**

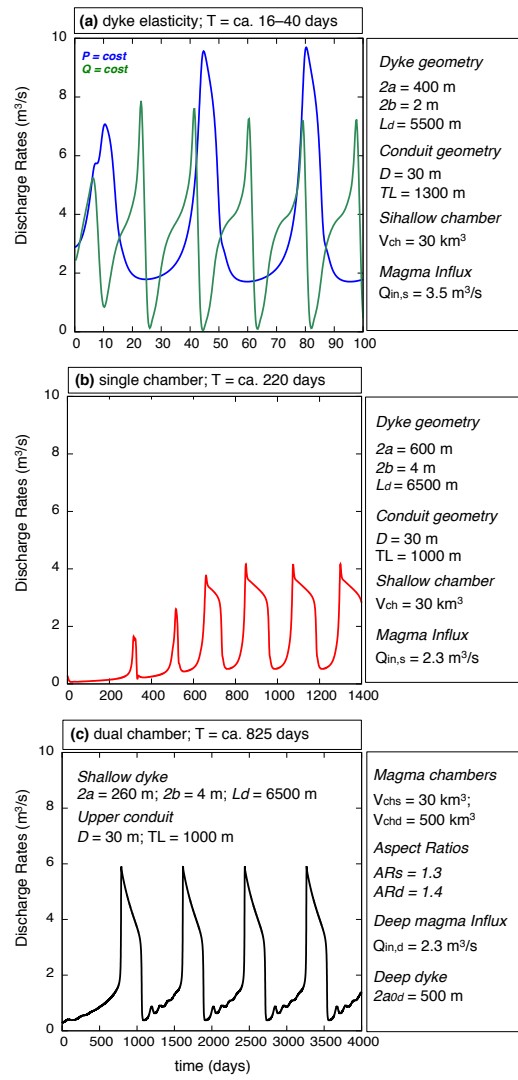








**Appendix A1**

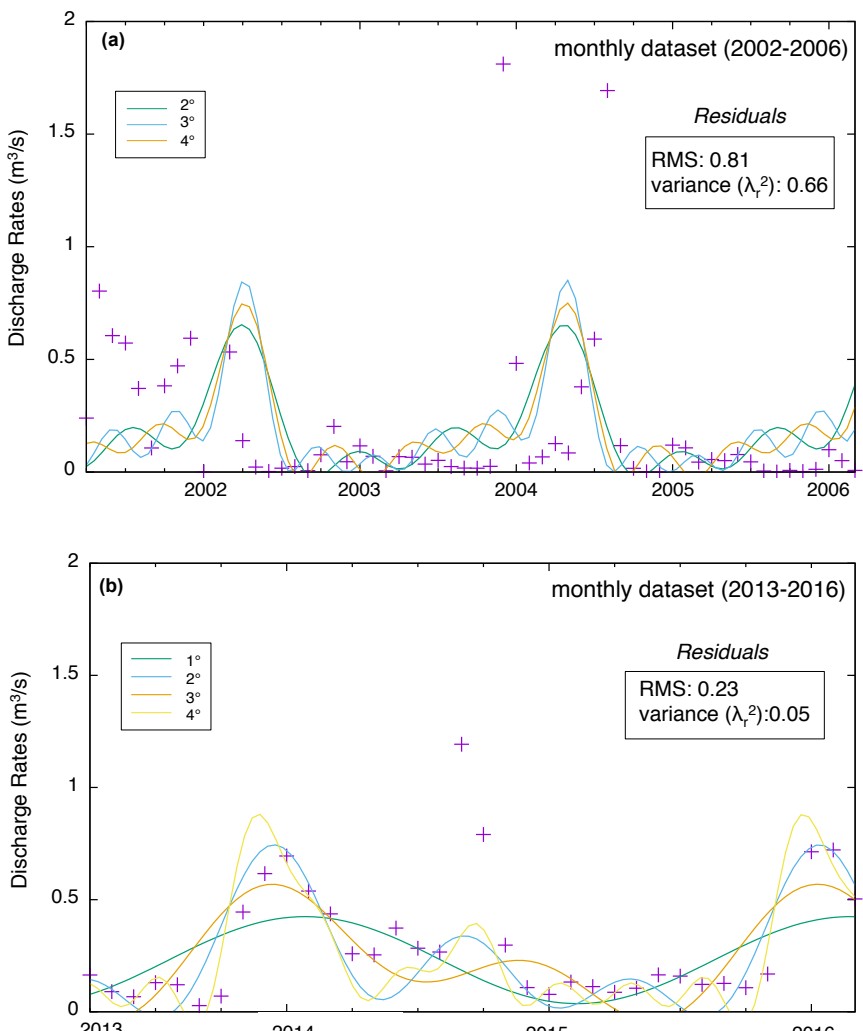





 **Appendix A2 – A3**

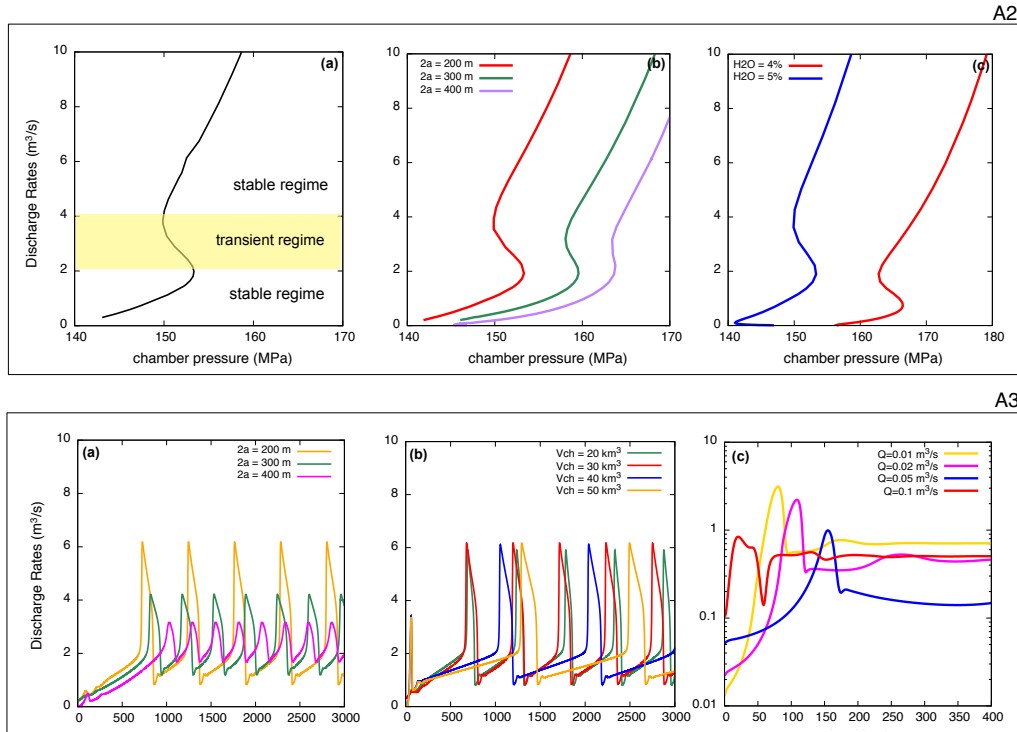

**Appendix A4**

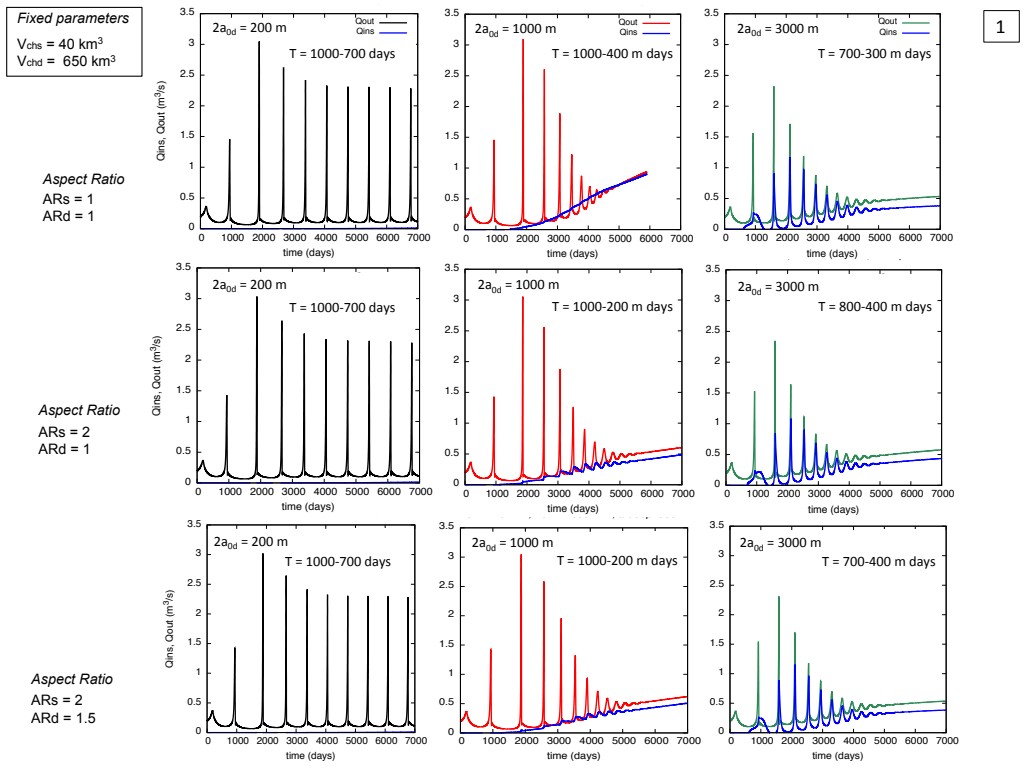



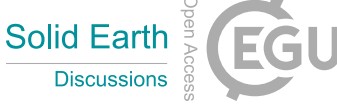



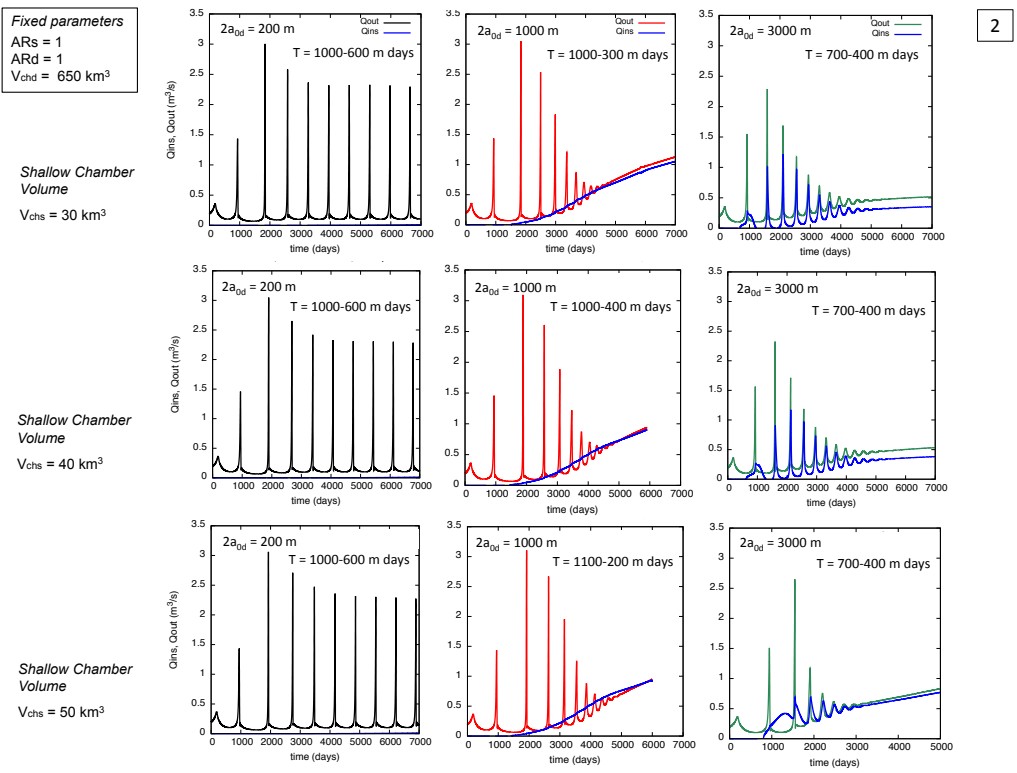




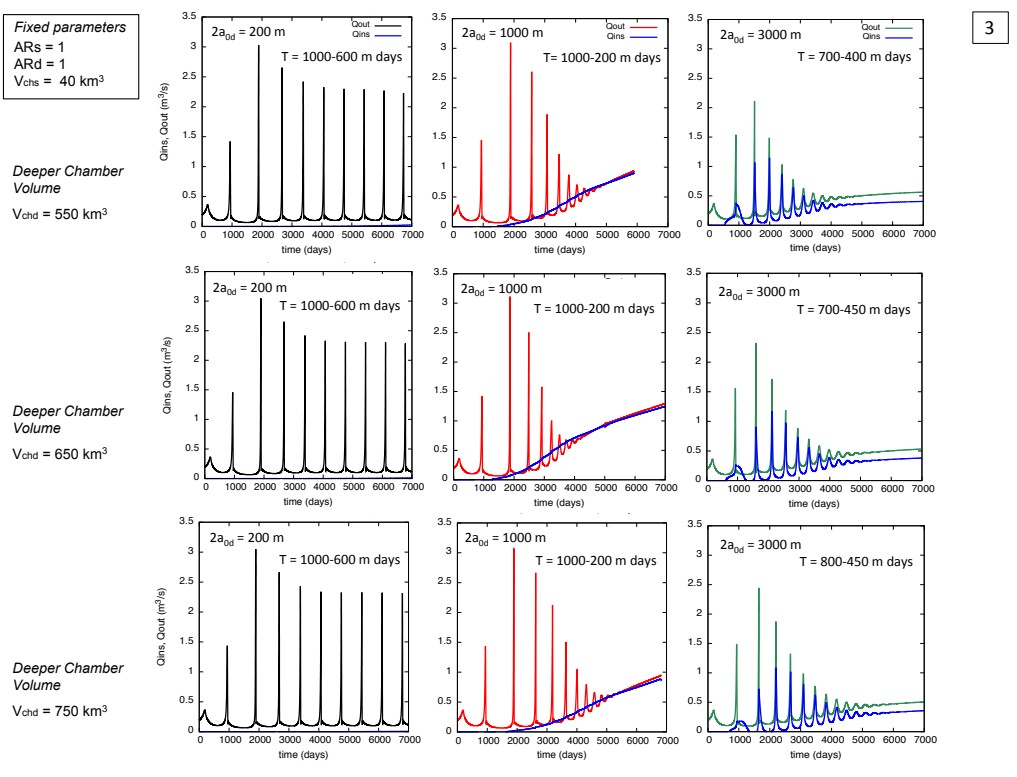





Fixed parameters : ARs = 1 ;  ARd = 1; Vchd = 650 km³ ; Vchs = 40 km³

- Qin_depth = 1  m³/s

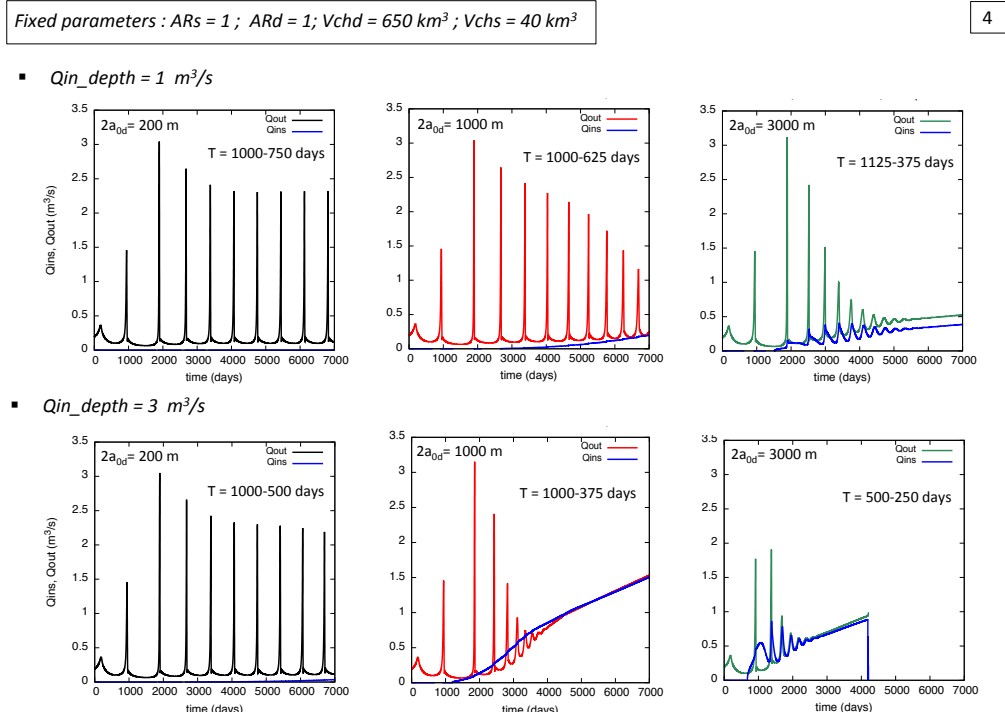

- Qin_depth = 3  m³/s
