# Peer review of "Cyclic activity of Fuego de Colima volcano (Mexico): insights from satellite thermal data and non-linear models"

_Solid Earth, 2019_

## Referee Comment (RC1) · Oleg Melnik (Referee) · 16 May 2019

The manuscript gives a detailed overview of the recent activity of Fuego de Colima volcano (Mexico). Several types of periodic behaviors are identified. Using wavelet analysis their periods are estimated. Numerical conduit flow model is used to explain observed periodicity. It is shown that different types of periodic signals are controlled by the different mechanisms - long term activity is better explained by dual magma chamber model, while the short period can result from the conduit processes. The problem is that all three timescales do not appear together in the model that accounts for dual magma chamber and to explain shorter timescales authors need to switch to

different settings. This approach should be discussed in more details. It is well known for Colima that Vulcanian explosions evacuate significant portions of the upper conduit and destroy the lava dome. The influence of these processes on at list short-term periodic regimes should be significant and requires some discussion in the paper.

---

## Referee Comment (RC2) · Anonymous Referee #2 · 30 May 2019

The authors report a cyclic activity of Fuego de Colima volcano (Mexico) in the period 1998-2018. Three periodicities have been identified by using wavelet analysis. Numerical simulations support the hypotheses that the cyclic behavior is due to the non-linear coupling between magma flow and crystallization in elastic dykes connecting one or two magma chambers with the surface. The used magma viscosity should be reported in the manuscript and/or in Table 1. In the figure Appendix A2-A3 (frame a), page 41, the yellow zone is indicated as the "transient regime"; I think that it is more appropriate to indicate that zone as "unstable" as described in the text. Line 394: it could be useful to report the units of T0 and T1 (perhaps months). The concentration of the dissolved gas reported in Table 1 (0.05-0.06 wt%) seems quite small for a magma that produces

vulcanian explosions (perhaps you mean weight fraction? please check).

---

## Author Comment (AC1) · 22 Jun 2019

We greatly appreciated the very interesting comments by the reviewer. As highlighted by the reviewer, our results indicate that long-term cyclicity is better explained by a dual magma chamber system, as previously highlighted by Melnik and Costa (2014). Short-term cyclicity can explained by the fluctuation of the shallow dyke, as previously highlighted by Costa et al. (2007-GRL). However it is very true that a model configuration only is not able to describe all the three periodicities investigated in our paper (long-, intermediate- and short-term). This is an actual numerical modelling limitation and probably in order to have a more sophisticated model able to describe all three time

scale at once it is necessary to incorporate more physics (e.g. full thermal effects) and consider fully 3D geometries. This would represent a great computational challenge but it is the direction where to go. In the revised version we discussed these limitations in the Discussion Section. We agree that the evacuation of significant portions of the upper conduit and the following destruction of the lava dome during Vulcanian explosions can affect periodicity. However, as it was shown by Costa et al. (2012) who considered the effect of 200 m plug collapse, such processes would mainly affect the very short-term periodic regimes and it should be more effective on sub-daily. Certainly, it is not excluded an exceptional large evacuation of the upper conduit would be able to influence longer periodicities (i.e. weekly – monthly) as those shown in our study but it is more likely affects sub-weekly periodicity that is not contemplated in this study due to the limitations of the observational dataset. This has been now discussed in the revised version in the Discussion Section. We hope this work could motivate the community towards the development of new 3D numerical models that should be able to describe the all the periodicity patterns described here in a more inclusive way. Please find attached the revised version of the manuscript with the marked changes.

On behalf of the authors, Sincerely, Silvia Massaro

Please also note the supplement to this comment:
https://www.solid-earth-discuss.net/se-2019-31/se-2019-31-AC1-supplement.pdf

**Supplement:**

[revised manuscript text omitted]

- *Qin_depth = 3  m³/s*

---

## Author Comment (AC2) · 22 Jun 2019

[revised manuscript text omitted]

- *Qin_depth = 3  m³/s*

---

## Author Response (AR1)

**Author's response**

**Reviewer#1 – Oleg Melnik**

"The manuscript gives a detailed overview of the recent activity of Fuego de Colima volcano (Mexico). Several types of periodic behaviors are identified. Using wavelet analysis their periods are estimated. Numerical conduit flow model is used to explain observed periodicity. It is shown that different types of periodic signals are controlled by the different mechanisms - long term activity is better explained by dual magma chamber model, while the short period can result from the conduit processes. The problem is that all three timescales do not appear together in the model that accounts for dual magma chamber and to explain shorter timescales authors need to switch to different settings. This approach should be discussed in more details. It is well known for Colima that Vulcanian explosions evacuate significant portions of the upper conduit and destroy the lava dome. The influence of these processes on at list short-term periodic regimes should be significant and requires some discussion in the paper."

*We greatly appreciated the very interesting comments by the reviewer. As highlighted by the reviewer, our results indicate that long-term cyclicity is better explained by a dual magma chamber system, as previously highlighted by Melnik and Costa (2014). Short-term cyclicity can explained by the fluctuation of the shallow dyke, as previously highlighted by Costa et al. (2007-GRL). However it is very true that a model configuration only is not able to describe all the three periodicities investigated in our paper (long-, intermediate- and short-term). This is an actual numerical modelling limitation and probably in order to have a more sophisticated model able to describe all three time scale at once it is necessary to incorporate more physics (e.g. full thermal effects) and consider fully 3D geometries. This would represent a great computational challenge but it is the direction where to go. In the revised version we discussed these limitations in the Discussion Section.*

*We agree that the evacuation of significant portions of the upper conduit and the following destruction of the lava dome during Vulcanian explosions can affect periodicity. However, as it was shown by Costa et al. (2012) who considered the effect of 200 m plug collapse, such processes would mainly affect the very short-term periodic regimes and it should be more effective on sub-daily. Certainly, it is not excluded an exceptional large evacuation of the upper conduit would be able to influence longer periodicities (i.e. weekly – monthly) as those shown in our study but it is*

*more likely affects sub-weekly periodicity that is not contemplated in this study due to the limitations of the observational dataset. This has been now discussed in the revised version in the Discussion Section.*

*We hope this work could motivate the community towards the development of new 3D numerical models that should be able to describe the all the periodicity patterns described here in a more inclusive way.*

**Reviewer#2 – Anonymous**

"The authors report a cyclic activity of Fuego de Colima volcano (Mexico) in the period 1998-2018. Three periodicities have been identified by using wavelet analysis. Numerical simulations support the hypotheses that the cyclic behavior is due to the non-linear coupling between magma flow and crystallization in elastic dykes connecting one or two magma chambers with the surface. The used magma viscosity should be reported in the manuscript and/or in Table 1. In the figure Appendix A2-A3 (frame a), page 41,the yellow zone is indicated as the "transient regime"; I think that it is more appropriate to indicate that zone as "unstable" as described in the text. Line 394: it could be useful to report the units of T0 and T1 (perhaps months). The concentration of the dissolved gas reported in Table 1 (0.05-0.06 wt%) seems quite small for a magma that produces Vulcanian explosions (perhaps you mean weight fraction? please check)."

*We thank the reviewer for the comments and corrections. In the revised version of the manuscript we reported the used magma viscosity value (see also Table 1) calculated internally by the numerical model. We made all corrections indicated: we changed the name "transient regime" into "unstable regime" in Appendix A2-A3 (frame a), and we corrected the value of concentration of the dissolved gas (5-6 wt%) in Table 1.*

On behalf of the authors
Sincerely,

*Silvia Massaro*

---

## Author Response (AR2)

**Author's response**

Dear Editor,

Thanks for your revision. We accepted the changes in the manuscript and we made also a better proofreading. We changed the Figure 3 adding the volcanological observations in order to easily understand the matches with the frequencies associated to periodicities.

On behalf of the authors
Sincerely,

*Silvia Massaro*

[revised manuscript text omitted]

L105

L106    **Fig. 2**

[Figure]

**Fig.3**

[Figure]

    **Fig. 4**

[Figure]

**Appendix A1-A2**

[Figure]

**Appendix A3**

[Figure]

[Figure]

Fixed parameters
ARs = 1
ARd = 1
V$_{chd}$ = 650 km³

Shallow Chamber
Volume

V$_{chs}$ = 30 km³

Shallow Chamber
Volume

V$_{chs}$ = 40 km³

Shallow Chamber
Volume

V$_{chs}$ = 50 km³

**Fixed parameters**
ARs = 1
ARd = 1
$V_{chs}$ = 40 km³

*Deeper Chamber Volume*

$V_{chd}$ = 550 km³

*Deeper Chamber Volume*

$V_{chd}$ = 650 km³

*Deeper Chamber Volume*

$V_{chd}$ = 750 km³

[Figure]

⌞138

*Fixed parameters : ARs = 1 ;  ARd = 1; Vchd = 650 km³ ; Vchs = 40 km³*

- *Qin_depth = 1  m³/s*

[Figure]

[Figure]

[Figure]

- *Qin_depth = 3  m³/s*

[Figure]

[Figure]

[Figure]

**Appendix A4**